# Lipidomic Analysis to Assess the Correlation between Ceramides, Stress Hyperglycemia, and HbA1c in Acute Myocardial Infarction

**DOI:** 10.3390/molecules28020716

**Published:** 2023-01-11

**Authors:** Melania Gaggini, Elena Michelucci, Rudina Ndreu, Silvia Rocchiccioli, Kyriazoula Chatzianagnostou, Sergio Berti, Cristina Vassalle

**Affiliations:** 1Institute of Clinical Physiology, National Research Council, Via G. Moruzzi 1, 56124 Pisa, Italy; 2Institute of Chemistry of Organometallic Compounds, National Research Council, Via G. Moruzzi 1, 56124 Pisa, Italy; 3Fondazione CNR-Regione Toscana G Monasterio, 56124 Pisa, Italy; 4Fondazione CNR-Regione Toscana G Monasterio, 54100 Massa, Italy

**Keywords:** ceramides, acute myocardial infarction, acute hyperglycemia, type 2 diabetes, glycated hemoglobin

## Abstract

Ceramides have been associated with cardiometabolic disease (e.g., acute myocardial infarction (AMI) and type 2 diabetes (T2D)) and adverse outcomes. Acute admission hyperglycemia (AH) is a transient glucose alteration in response to stress. As glycated hemoglobin (HbA1c) reflects the glycemia over a longer period of time, its use may be helpful in distinguishing between the AH and hyperglycemia associated with T2D in the AMI setting. The aim was to assess the correlation of ceramides with both AH (defined as an admission glucose level ≥140 mg/dL in the absence of T2D) and HbA1c-T2D and other demographic, clinical, and inflammatory-related biomarkers in AMI. High-performance liquid chromatography–tandem mass spectrometry was used to identify nine ceramide species, and their three ratios, in 140 AMI patients (FTGM coronary unit, Massa, Italy). The ceramides did not correlate with stress hyperglycemia, but specific species were elevated in T2D-AMI. Moreover, some ceramides were associated with other cardiometabolic risk factors. Ceramides assessment may be helpful in better understanding the pathogenic molecular mechanisms underlying myocardial acute events and cardiometabolic risk, as a basis for the future evaluation of their role as prognostic predictors and therapeutic targets in T2D-AMI patients.

## 1. Introduction

Ceramides (Cer) are the basic sphingolipid units produced by six fatty acyl selective ceramide synthases, starting from fatty acids and serine or sphingomyelin, and they are the regulators of numerous cellular pathways (e.g., those involved in cellular differentiation, apoptosis and proliferation, and the activation of a variety of signalling cascades) [1]. Circulating ceramides have been proposed as markers of several inflammatory/oxidative stress responses, cardiometabolic risk, such as acute myocardial infarction (AMI) and type 2 diabetes (T2D), and related adverse outcomes [2,3,4]. We previously evaluated the relationship of three ceramide species (18:1/16:0, 18:1/18:0, and 18:1/24:1) and their ratios with CV risk factors, inflammatory parameters, and left ventricular function in AMI, evidencing the association between specific ceramide species and CV risk, inflammation, and disease severity [5].

The alterations of blood glucose in patients with AMI may reflect a temporary stress-induced phenomenon rather than a previous status of dysglycemia caused by pre-existing diabetes, although both conditions are associated with adverse outcomes and share different characteristics, including increased oxidative stress and inflammation [5,6,7,8]. Specifically, elevated admission hyperglycemia (AH) has been reported to negatively affect AMI outcome as AH results are associated with a higher risk of all-cause mortality and complications (e.g., heart failure and cardiac shock) [6,7,8]. As glycated hemoglobin (HbA1c) reflects the average level of glycemia over a longer period of time (over 2 to 3 months), its use may be helpful in distinguishing between stress hyperglycemia and hyperglycemia associated with T2D in this acute setting [9].

The aim of this study was to assess the correlation of nine ceramide species with both AH in the absence of T2D (reflecting a stress-related response) and with HbA1c/T2D (reflecting a real chronic metabolic alteration in AMI patients) and with other demographic and clinical factors in AMI patients.

## 2. Results

A total of 140 patients with AMI admitted to the Heart Hospital Pasquinucci of Fondazione Gabriele Monasterio were enrolled. The patients were classified into three groups based on their history of diabetes and their plasma glucose/HbA1c levels on admission:

Group 1 (*n* = 102 [73%]): No-diabetic patients without AH, noT2D.

Group 2 (*n* = 12 [9%]): No-diabetic patients with AH, AH.

Group 3 (*n* = 26 [18%]): Diabetic patients, T2D.

The demographic and clinical characteristics of the studied population are reported in Table 1. There were fewer patients reporting a smoking history in the AH and T2D groups and, as expected, a lower percentage of obese subjects in the noT2D group.

Figure 1 showed higher levels of Cer(d18:1/18:0) and Cer(d18:1/18:0)/Cer(d18:1/24:0) in T2D patients but not in those with AH or in the noT2D patients (0.41 ± 0.29, 0.31 ± 0.09, and 0.31 ± 0.16 µM, *p* = 0.046; 0.14 ± 0.08, 0.114 ± 0.03, and 0.11 ± 0.05 µM, *p* = 0.044, respectively).

All the ceramides correlated with each other, although with different relationship strengths, as reported in Table 2.

The correlation between the ceramides and the CV risk factors is reported in Table 3. In particular ceramides, 18:1/16:0, 18:1/18:0, and 18:1/20:0, (18:1/16:0)/(18:1/24:0), (18:1/18:0)/(18:1/24:0), and (18:1/24:1)/(18:1/24:0) correlated significantly with gender and 18:1/16:0, and the ratios of the ceramides also significantly correlated with age. Moreover (18:1/18:0)/(18:1/24:0) resulted in being significantly associated with dyslipidemia.

When the effect of the therapy was specifically evaluated, Cer(d18:1/22:0), Cer(d18:1/23:0), Cer(d18:1/24:0), and Cer(d18:2/22:0)) resulted significantly reduced in patients taking lipid lowering therapy (mainly statins), while antiplatelet therapy (mainly aspirin) was associated with a reduction in Cer(d18:1/24:0) and Cer(d18:1/25:0) (Figure 2). However, we did not observe any significant association between antihypertensive (diuretics, calcium channel blockers, ACE inhibitors, and beta-blockers) and antidiabetic therapy (insulin or oral hypoglycemic therapy) and ceramide levels.

## 3. Discussion

In the present study, we evaluated nine ceramide species and their three ratios, and for the first time, we evaluated the association of ceramide species with AH beyond T2D. We observed that the ceramides did not correlate with AH, but specific species resulted in being higher in T2D-AMI when compared with the noT2D-AMI patients. Moreover, the ceramides were associated with cardiometabolic risk factors in AMI, and the levels of specific species decreased according to some CV drugs.

The assessment of distinct ceramide species by using a high-throughput methodology (e.g., LC-MS/MS) is now profitable and efficacious, suggesting the possibility of introducing ceramides in the clinical practice after further evaluation of their clinical significance as potential biomarkers in cardiometabolic risk and disease [10,11]. In fact, the specific and complex relationships of different chains of fatty acids in ceramides with different pathophysiological conditions clearly demonstrates the complexity of these biological pathways, which require the simultaneous evaluation of increased, decreased, or unchanged specific species [10,11]. Accordingly, specific ceramide classes have been found to be modulated in many pathological conditions, including atherosclerosis. In particular, it has been supposed that ceramides have a causal role in coronary artery disease [2] and can predict death and major adverse cardiovascular events (MACE) in these patients, with Cer(d18:1/16:0), Cer(d18:1/18:0), Cer(d18:1/24:1), and Cer(d18:1/24:0), among all the ceramide species, being those with the closest association with events [12].

Moreover, a close relationship has been found between distinct serum ceramides and other sphingolipids and T2D [10,13]. In fact, an increase in the total and specific ceramide plasma concentration leads to the promotion of insulin resistance (e.g., by activating intracellular inflammatory pathways and increasing cytokine production by macrophage through Toll-like receptor 4 signalling), whereas LDL-ceramide tended to correlate negatively with insulin sensitivity [10,14]. Our results, which show an elevation in Cer(d18:1/18:0) and Cer(d18:1/18:0)/Cer(d18:1/24:0) in AMI-T2D patients, are in agreement with the previous results [15,16]. Specifically, Cer(d18:1/18:0) and its ratio to Cer(d18:1/24:0) have recently been found able to accurately predict acute coronary syndrome-T2D, confirming the previous evidence which suggests that Cer(d18:1/18:0) is particularly adverse for insulin sensitivity and promotes adipose inflammation [15,16]. This may suggest that Cer(d18:1/18:0) and its ratio to Cer(d18:1/24:0) could represent a new biomarker to be applied in the AMI-T2D clinical setting.

On the other hand, admission hyperglycemia (AH) is a transient condition, representing a significant prognostic determinant of in-hospital complications and mortality in acute diseases, including AMI, as a result of an inflammatory and adrenergic response to ischemic stress [17,18]. In fact, a recent meta-analysis showed that AMI patients with new hyperglycemia had a 3.6-fold increased risk of mortality during hospitalization in comparison with those who were normoglycemic [18]. Moreover, AMI patients with elevated glucose and non-elevated HbA1c have the highest mortality and major adverse cardiovascular events due to the greater risk of incurring cardiogenic shock and reinfarction at 30 days [19]. Thus, the research of biomarkers which may correlate with acute hyperglycemia and explain the pathological mechanisms underlying this condition are of interest [20,21]. However, the lack of correlation between ceramides and AH suggests that these biomarkers do not play a role in this condition, despite the fact that hyperglycemia also appears to be related to insulin resistance, increased oxidative stress and inflammation, and a prothrombotic state [18]. Perhaps the transient nature of this condition may hinder the development of this relationship.

Elevated levels of ceramides are also related to different cardiovascular risk factors, including dyslipidemia [22], as we found in our population. In particular, in dyslipidemia, the extra fat supply to tissues, which is not destined for lipid storage, drives lipotoxicity and cellular dysfunction, which underlie cardiometabolic disease [23,24]. Among the number of lipids produced under these conditions, sphingolipids (e.g., ceramides or their metabolites) are among the most dangerous because they affect insulin sensitivity, pancreatic β cell function, vascular reactivity, and mitochondrial metabolism [23,24].

Ceramides may be the target of therapeutical interventions in coronary artery disease patients. Accordingly, statins, which are effective for the reduction in the coronary artery disease risk and for making the plaques more stable, are also found to reduce some ceramide species (Cer(d18:1/16:0), Cer(d18:1/18:0), Cer(d18:1/24:1), and Cer(d18:1/24:0)) [25,26,27,28].

Accordingly, we also observed that Cer(d18:1/22:0), Cer(d18:1/23:0), Cer(d18:1/24:0), and Cer(d18:2/22:0)) were significantly reduced in patients taking lipid lowering therapy (mainly statins). Interestingly, the inhibition of ceramide biosynthesis in experimental models improves hypertriglyceridemia, evidencing the possibility of therapeutic interventions directed to ceramide reduction as treatments for cardiometabolic disease (e.g., myriocin, an irreversible and high-affinity inhibitor of serine palmitoyltransferase, the enzyme involved in the ceramide de novo synthesis in the endoplasmic reticulum) [10,29,30,31,32]. Moreover, although we did not observe any significant effects of the antihypertensive and antidiabetic therapy on ceramide levels, antiplatelet therapy (mainly aspirin) was associated with a reduction in Cer(d18:1/24:0) and Cer(d18:1/25:0). Some previous data suggested that aspirin may reduce the sphingosine-1-phosphate (S1P) and sphinganine-1-phosphate (SA1P) levels in the plasma of healthy volunteers (a group at one week/dose of 75 mg and another group who received one 300 mg dose of the drug) [33]. However, in this case, as S1P is a potent cardio-protectant, the reduction in its concentration after the loading dose of aspirin could be an undesired side effect. In any case, although this study was not designed for this scope, these results may be the basis-along with the other available data-to deepen the significance of the effects of cardiovascular drugs on ceramides, specifically focusing on the type of drugs, the duration of the therapy, and the doses. Moreover, it is important to further investigate whether each ceramide species elevation represents a direct step in the causal pathway of a disease or whether the increase observed is just a counter reaction to an adverse event. In this context, the effects associated with the cardiovascular drugs must be evaluated specifically for each ceramide specie because their chain length may affect the different functions of the ceramides and their effects (beneficial vs. adverse) on the cardiometabolic risk (e.g., in relation to the therapy, we find reduced long-chain ceramides, which seem to retain a beneficial effect on the CV risk) [34].

## 4. Materials and Methods

### 4.1. Population Characteristics and Data Acquisition

ST-segment elevation myocardial infarction (STEMI) patients were enrolled at the Ospedale del Cuore G. Pasquinucci-Clinical Cardiology Department (Massa, Italy). The STEMI definition follows the published SC/ACCF/AHA/WHF guidelines for STEMI criteria and management [35].

Within 90 min of admission, all the patients underwent coronary angiography with subsequent PCI. The data concerning all the information of the subjects, including demographic and laboratory parameters, cardiovascular history, and cardiovascular risk factors, were extracted from the computerized clinical database (MATRIX) of our hospital. Hypertension was defined in the case of blood pressure higher than 140/90 mmHg or the current use of antihypertensive medications. Dyslipidemia was defined by the use of lipid-lowering treatments or by fasting low-density lipoprotein levels >150 mg/dL. The admission plasma glucose of 140 mg/dl (7.8 mmol/L) and above was chosen to identify patients with acute phase hyperglycemia (AH) [36]. Type 2 diabetes (T2D) was defined as the use of antidiabetic treatment, a fasting glucose >126 mg/dl (7 mmol/l) on two separate tests prior to the the acute event, or the finding of HbA1c >6.49% at AMI admission.

The patients were considered eligible to be enrolled in the study based on the following inclusion criteria: (1) male and female adult patients, admitted to the coronary care unit for STEMI and (2) patients subject to percutaneous coronary revascularization and stenting of the culprit lesion within 24 h from the onset of symptoms. The exclusion criteria were: (1) severe systemic diseases; (2) systemic inflammatory disease; (3) patients refusing or unable to supply written informed consent; and (4) patients who had not undergone HbA1c. Standard therapy was administered to all eligible patients. Informed consent was obtained from each patient (or from their relatives where necessary) before the angiogram, and the study was approved by the local ethics committee (number 19214, 11 February 2021).

### 4.2. Plasma Processing and HPLC-MS/MS Analysis

For ceramides identification, we analysed 140 samples, stored at −80 °C. The lipid extraction and HPLC-MS/MS setting are shown in Figure 3. The detailed analytical procedure, the chemical standard and material, are described in Appendix A.

### 4.3. Statistical Analysis

The continuous variables were reported as mean ± SD and the categorical variables as numbers (percentages). The calculation of the differences between the categorical variables was performed using chi-square analysis. The *t*-test was used to compare the means of the two ceramide groups according to the presence/absence of CV risk factors. Comparisons among variables of three or more independent groups were analysed using ANOVA and the Scheffè test, which is used for post hoc analysis. Regression analysis was performed to assess the relationship between the ceramide species. A *p*-value of <0.05 was chosen as the level of significance.

## 5. Limitations of the Study

This was a single-center study with a small sample size; thus, the data must be confirmed in further trials.

## 6. Conclusions

In our population of AMI patients, ceramides were associated with T2D and cardiometabolic risk factors but not with AH. Their levels are associated with other CV risk factors and are affected by the CV therapy. Thus, further evaluation of ceramides may be helpful in better understanding the pathogenic molecular mechanisms underlying myocardial acute events and cardiometabolic risk, as well as their role as potential targets of therapeutical strategies, especially in those with T2D who incur a AMI.

## Figures and Tables

**Figure 1 molecules-28-00716-f001:**
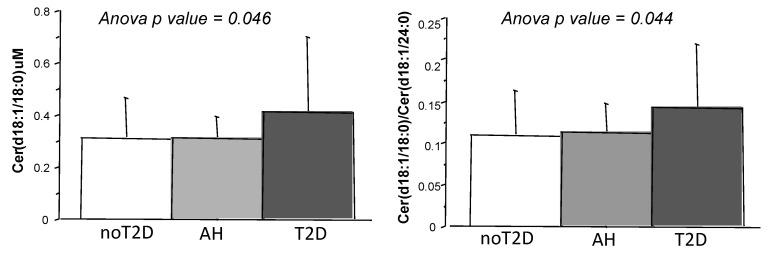
Levels of Cer(d18:1/18:0) and Cer(d18:1/18:0)/Cer(d18:1/24:0) in noT2D, AH, and T2D patients.

**Figure 2 molecules-28-00716-f002:**
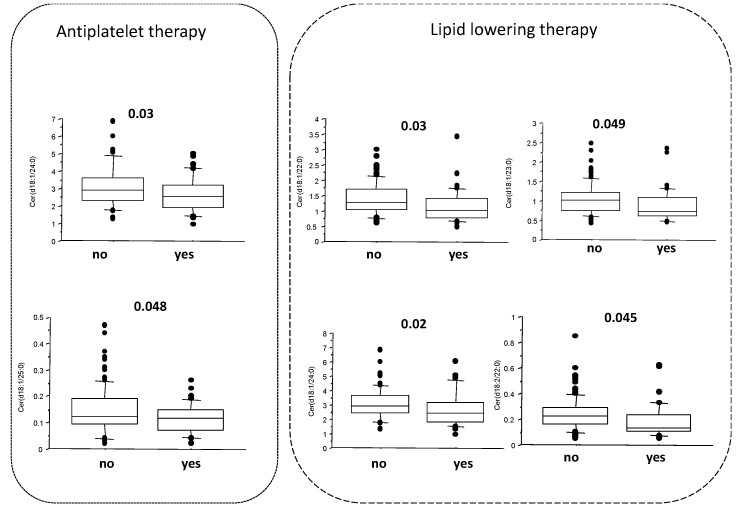
Reduction in specific ceramide species according to antiplatelet and lipid lowering therapy.

**Figure 3 molecules-28-00716-f003:**
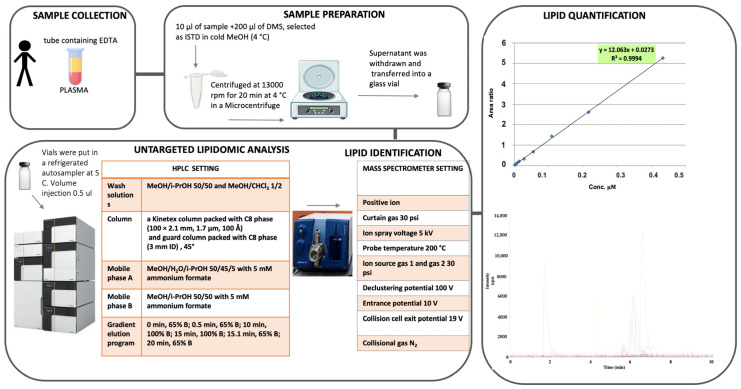
Plasma processing and HPLC-MS/MS setting and analysis.

**Table 1 molecules-28-00716-t001:** Demographic, clinical, and laboratory parameters in the 140 AMI patients.

Variable	Overall Population *n* = 140	noT2D *n* = 102	AH *n* = 12	T2D *n* = 26	*p* Value (χ^2^ Test)
Females	37 (26)	23 (22)	4 (33)	10 (38)	0.52
Age (>69 years 50th percentile)	63 (45)	40 (35)	7 (58)	16 (61)	0.38
Hypertension	88 (63)	59 (58)	7 (58)	22 (81)	0.18
Dyslipidemia	69 (49)	45 (44)	7 (58)	17 (65)	0.53
Smoking habit	33 (24)	28 (27)	2 (17)	3 (11)	0.06
Obesity	23 (16)	9 (9)	4 (33)	10 (38)	0.048

AH defined for admission with hyperglycemia (140 mg/dL–7.8 mmol/L and above). χ^2^ test for three categories (NoT2D, AH, T2D) was performed for each variable reported in the table (female or male sexes, age < or > 69 years, presence or not of hypertension, dyslipidemia, smoking habit, or obesity).

**Table 2 molecules-28-00716-t002:** Correlation between ceramides.

Ceramide	18:1/18:0	18:1/20:0	18:1/22:0	18:1/23:0	18:1/24:0	18:1/24:1	18:1/25:0	18:2/22:0
**18:1/16:0**	0.8 <0.001	0.85 <0.001	0.71 <0.001	0.62 <0.001	0.54 <0.001	0.83 <0.001	0.51 <0.001	0.6 <0.001
**18:1/18:0**		0.91 <0.001	0.72 <0.001	0.62 <0.001	0.46 <0.001	0.8 <0.001	0.4 <0.001	0.57 <0.001
**18:1/20:0**			0.86 <0.001	0.77 <0.001	0.64 <0.001	0.88 <0.001	0.6 <0.001	0.6 <0.001
**18:1/22:0**				0.9 <0.001	0.88 <0.001	0.82 <0.001	0.65 <0.001	0.91 <0.001
**18:1/23:0**					0.84 <0.001	0.77 <0.001	0.76 <0.001	0.88 <0.001
**18:1/24:0**						0.63 <0.001	0.74 <0.001	0.77 <0.001
**18:1/24:1**							0.63 <0.001	0.7 <0.001
**18:1/25:0**								0.53 <0.001

Data are reported as r, *p* value.

**Table 3 molecules-28-00716-t003:** Correlation between CV risk factor and ceramides.

Parameters	Ceramide Species											
	18:1/16:0	18:1/18:0	18:1/20:0	18:1/22:0	18:1/23:0	18:1/24:0	18:1/24:1	18:1/25:0	18:2/22:0	18:1/16:0 18:1/24:0	18:1/18:0 18:1/24:0	18:1/24:1 18:1/24:0
Females vs. males	1.01 (0.38) vs. 0.89 (0.25) = 0.03	0.4 (0.26) vs. 0.3 (0.146) = 0.005	0.255 (0.146) vs. 0.21 (0.09) = 0.042	1.36 (0.6) vs. 1.29 (0.52) ns	1 (0.44) vs. 1 (0.41) ns	2.8 ( 1.1) vs. 3 (1.2) ns	1.51 (0.7) vs. 1.34 (0.51) ns	0.14 (0.09) vs. 0.13 (0.08) ns	0.25 (0.13) vs. 0.22 (0.14) ns	0.38 (0.127) vs. 0.323 (0.120) = 0.018	0.144 (0.066) vs. 0.105 (0.051) < 0.001	0.548 (0.174) vs. 0.475 (0.0165) = 0.027
Age < 69 (50th percentile) vs. >69	0.86 (0.25) vs. 0.99 (0.33) = 0.016	0.31 (0.16) vs. 0.35 (0.22) ns	0.22 (0.1) vs. 0.23 (0.13) ns	1.34 (0.52) vs. 1.28 (0.56) ns	1 (0.4) vs. 0.97 (0.43) ns	3.14 (1.1) vs. 2.77 (1.1) ns	1.30 (0.5) vs. 1.48 (0.63) ns	0.136 (0.09) vs. 0.132 (0.08) ns	0.22 (0.13) vs. 0.23 (0.14) ns	0.288 (0.08) vs. 0.39 (0.14) < 0.001	0.1 (0.049) vs. 0.13 (0.063) = 0.004	0.428 (0.116) vs. 0.563 (0.188) < 0.001
No hypertension vs. hypertension	0.9 (0.16) vs. 0.94 (0.34) ns	0.3 (0.13) vs. 0.34 (0.22) ns	0.21 (0.08) vs. 0.23 (0.12) ns	1.22 (0.4) vs. 1.3 (0.6) ns	0.9 (0.28) vs. 1.01 (0.46) ns	2.8 (0.9) vs. 3.0 (1.2) ns	1.28 (0.29) vs. 1.43 (0.65) ns	0.12 (0.06) vs. 0.14 (0.09) ns	0.21 (0.1) vs. 0.23 (0.15) ns	0.356 (0.13) vs. 0.335 (0.12) ns	0.121 (0.06) vs. 0.115 (0.06) ns	0.51 (0.19) vs. 0.49 (0.16) ns
No dyslipidemia vs. dyslipidemia	0.88 (0.27) vs. 0.96 (0.32) ns	0.29 (0.16) vs. 0.36 (0.2) ns	0.2 (0.1) vs. 0.24 (0.12) ns	1.26 (0.5) vs. 1.34 (0.6) ns	0.96 (0.4) vs. 1.0 (0.4) ns	2.9 (1.1) vs. 3.0 (1.2) ns	1.35 (0.6) vs. 1.42 (0.6) ns	0.14 (0.09) vs. 0.13 (0.08) ns	0.205 (0.11) vs. 0.239 (0.15) ns	0.325 (0.121) vs. 0.353 (0.127) ns	0.105 (0.049) vs. 0.126 (0.063) = 0.04	0.48 (0.18) vs. 0.51 (0.16) ns
No smoking habit vs. smoking habit	0.93 (0.33) vs. 0.91 (0.21) ns	0.33 (0.21) vs. 0.32 (0.14) ns	0.22 (0.12) vs. 0.23 (0.08) ns	1.29 (0.6) vs. 1.35 (0.4) ns	1 (0.5) vs. 1 (0.3) ns	2.9 (1.1) vs. 3.2 (1.2) ns	1.41 (0.63) vs. 1.33 (0.38) ns	0.13 (0.1) vs. 0.13 (0.07) ns	0.23 (0.15) vs. 0.21 (0.1) ns	0.353 (0.13) vs. 0.311 (0.1) ns	0.12 (0.06) vs. 0.11 (0.05) ns	0.52 (0.17) vs. 0.45 (0.15) ns

*p* value (*t* test for unpaired data), vs. = versus, ns = not significant.

## Data Availability

The data that support the findings of this study are available upon reasonable request (e.g., research purpose) from the authors.

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
