# Peer review of "Lipidomic Analysis to Assess the Correlation between Ceramides, Stress Hyperglycemia, and HbA1c in Acute Myocardial Infarction"

_molecules, 2023, doi:10.3390/molecules28020716_

Round 1

Reviewer 1 Report

The manuscript by Gaggini et al. entitled “Lipidomic analysis to assess the correlation between ceramides, stress-hyperglycemia and HbA1c in acute myocardial infarction” conducts a throughout assessment of the relationship between ceramides and the presence of acute hyperglycemia (AH) with and without type 2 diabetes (T2D) in the setting of acute myocardial infarction (AMI). Overall, they found that specific ceramide species associated to T2D, other cardiometabolic risk factors as well as cardiovascular therapy (with antiplatelet agents and statins) but not to AH. Despite the small sample size, the manuscript is well-performed and offers new therapeutic and diagnostic avenues for patients with T2D developing an AMI. I only have minor suggestions:

-          What was the rationale for selecting the nine ceramide species?

-          Some sentences require grammatical revision (e.g., line 18, 46, etc.)

-          Line 60 (page 2) – …Table 1.

-          In Table 1, to what comparison refers the P-value? NoT2D vs AH, AH vs T2D, NoT2D vs T2D, ...?

-          In Figure 1, standard deviation or standard error of the mean should be indicated with error bars.

Reviewer 2 Report

The study found association of specific ceramide levels with type 2 diabetes and cardiometabolic risk factors in type 2D-AMI patients. This could indicate that such changes in specific ceramide levels (or ratios) may have prognistic value if further studies could confirm this in larger cohorts.

There are several typo and grammar errors; the authros should carefully check the text.

Error bars showed be added to both panels in Figure 1.

Table 3, first line, last column: 8:1/24:0 should be 18:1/24:0
